# Reutilization of Silicon-Cutting Waste via Constructing Multilayer Si@SiO_2_@C Composites as Anode Materials for Li-Ion Batteries

**DOI:** 10.3390/nano14070625

**Published:** 2024-04-02

**Authors:** Yi Sun, Jingyi Wu, Xingjie Chen, Chunyan Lai

**Affiliations:** Shanghai Key Laboratory of Materials Protection and Advanced Materials in Electric Power, Shanghai University of Electric Power, Shanghai 200090, China; 15797926318@163.com (Y.S.); wujingyi0728@163.com (J.W.); a1317358861@163.com (X.C.)

**Keywords:** silicon-cutting waste, reutilization, silicon/carbon anode, scalable synthesis, lithium-ion batteries

## Abstract

The rapid development of the photovoltaic industry has also brought some economic losses and environmental problems due to the waste generated during silicon ingot cutting. This study introduces an effective and facile method to reutilize silicon-cutting waste by constructing a multilayer Si@SiO_2_@C composite for Li-ion batteries via two-step annealing. The double-layer structure of the resultant composite alleviates the severe volume changes of silicon effectively, and the surrounding slightly graphitic carbon, known for its high conductivity and mechanical strength, tightly envelops the silicon nanoflakes, facilitates ion and electron transport and maintains electrode structural integrity throughout repeated charge/discharge cycles. With an optimization of the carbon content, the initial coulombic efficiency (ICE) was improved from 53% to 84%. The refined Si@SiO_2_@C anode exhibits outstanding cycling stability (711.4 mAh g^−1^ after 500 cycles) and rate performance (973.5 mAh g^−1^ at 2 C). This research presents a direct and cost-efficient strategy for transforming photovoltaic silicon-cutting waste into high-energy-density lithium-ion battery (LIB) anode materials.

## 1. Introduction

The increasing need for sustainable energy sources, driven by energy shortages and environmental concerns, has accelerated the progress of renewable energy technologies [1]. Silicon-based photovoltaic technology stands out as a cost-effective solution for power generation [2]. Nonetheless, the photovoltaic industry of China faces a substantial issue with the production of silicon-cutting waste: approximately 35% of solar-grade silicon ingots become waste, leading to annual losses exceeding USD 2.8 billion [3,4]. Additionally, the silicon nanoflakes, about 1 μm in particle size, pose significant environmental risks by contaminating water and soil [5]. For this reason, the efficient recycling and utilization of silicon-cutting waste are imperative.

Various methods have been developed to recycle silicon-cutting waste, such as developing silicon alloys [5], manufacturing silicon nitride [6], and creating surface-modified adsorbents [7]. Utilizing silicon-based materials with ultrahigh theoretical specific capacity as lithium-ion battery (LIB) electrode material is also an efficient pathway for silicon-cutting waste recovery [8]. However, silicon-based anodes encounter issues such as the isolation of active materials and copper foils, the fragmentation of electrodes, and the instability of solid electrolyte interphase (SEI) layers, causing rapid electrolyte depletion [9,10] due to their volume expansion rate of over 300% during cycles and the poor electrical conductivity [11,12]. Therefore, numerous challenges and obstacles impede the progress and effective application of silicon-based anodes.

To address these issues mentioned above, which could lead to inferior performance, various strategies have been investigated in recent years [13,14,15,16,17,18,19,20,21,22,23,24]. The use of nanoscale silicon particles is noteworthy, as such use reduces volume changes, shortens ion transport distances, and increases contact area with the electrolyte [13]. Nevertheless, their electrochemical performance is constrained by disadvantages like repeated SEI film formation, low tap density and severe agglomeration [14]. Integrating carbonaceous matrices like graphitic carbon [15,16], amorphous carbon [17,18], graphene [19,20], graphene oxide [21,22], carbon nanotubes [23,24,25], carbon nanofibers [26,27] and carbon cloth [28], has been demonstrated to be a highly efficacious approach to improve the mechanical flexibility, SEI stability, and electrical conductivity of silicon anode. Graphitic carbon, in particular, enhances the electroconductivity and Li^+^ transport performance of silicon nanoparticles. Nonetheless, synthesizing silicon–graphitic carbon composites often involves challenging conditions, complex processes, and expensive materials [17].

Based on the recycling of silicon-cutting waste and the demand for high-performance LIB anodes, a straightforward and economical method utilizing silicon-cutting waste and coal tar pitch to fabricate a multilayered Si@SiO_2_@C composites as anodes for high-performance LIBs was developed. The optimized composite features a multilayer structure that accommodates volume changes of silicon and enhances Li^+^ diffusion rates. Additionally, enveloping silicon nanoflakes with slightly graphitic carbon forms a highly conductive layer that not only preserves the structural stability of the electrode but also facilitates ion/electron transport in repetitive cycles. Owing to the distinctive structure of the composite, the electrode demonstrates exceptional rate performance, extended cycling life, and superior coulombic efficiency (CE). The Si@SiO_2_@C electrode also exhibits outstanding cycling performance and high energy density in the full-cell test, which demonstrates the potential of composites in practical applications. Furthermore, the superiorities in method and performance are proved by comparing this work to the state of art. Significantly, this research not only offers a simple and economical solution for the reutilization of the waste from silicon ingot cutting but also shows potential to advance the sustainable resource utilization of silicon-cutting waste.

## 2. Material and Methods

### 2.1. Synthesis of Composites

**Pretreatment of silicon-cutting waste (SCW).** The powder of silicon-cutting waste agglomerations (5 g) and deionized water (100 mL) were first put into a container of high-speed homogenizer and stirred at 10,000 rpm for 90 min, then dried to a constant weight.

**Synthesis of Si@SiO_2_.** The preprocessed SCW powder was heated to a temperature of 600 °C for a duration of 2 h in an air atmosphere to oxidize the surface silicon and remove the organic residuals.

**Synthesis of multilayered Si@SiO_2_@C.** The prepared Si@SiO_2_ powder (0.9 g) and pitch (0.3 g, Cogermin Co, Essen, Germany) were dispersed into a tetrahydrofuran solution (50 mL, THF, AR, 99.5%, Sigma-Aldrich, St. Louis, MI, USA) and mixed at 400 rpm until the THF was evaporated entirely. The resulting mixture was then subjected to vacuum drying at a temperature of 40 °C for a duration of 2 h. Subsequently, the desiccated powder was carbonized in a nitrogen atmosphere (200 mL min^−1^), heated from 25 to 450 °C at 5 °C min^−1^ and annealed at 450 °C for 2 h, and thence heated to 900 °C at 5 °C min^−1^ and kept for 2 h. The product was labeled as Si@SiO_2_@C-2, 3 and 4, respectively, based on the different addition of Si@SiO_2_ and pitch (2:1, 3:1 and 4:1 by mass).

### 2.2. Characterization of Material

A scanning electron microscope (SEM, JSM-7800F, Tokyo, Japan) was employed to observe the general surface morphologies of composites. A transmission electron microscope (TEM, JEM-2100F, Tokyo, Japan) was used to measure the double-layered structure of the morphology. The crystal phases of composites were collected by X-ray diffraction (XRD, Bruker D8, Berlin, Germany). Thermogravimetric analysis (TGA, TGA/DSC 3+, from 30 °C to 800 °C, 10 °C min^−1^) was employed to estimate the carbon content of synthesized composites. Raman spectroscopy (BX41 with a laser of 532 nm) was used to analyze the graphitization degree. An X-ray fluorescence spectrometer (XRF, TM-4200, Waltham, USA) was employed to analyze the amount of all the elements in the SCW. The valence states and chemical bonds of surface elements were investigated by X-ray photoelectron spectroscopy (XPS, K-α Nexsa, Waltham, USA). A nitrogen adsorption–desorption analyzer (ASAP 2460, Georgia, USA) was used to determine the pore property. The Brunauer–Emmett-Teller analysis (BET) and Barrett–Joyner–Halenda (BJH) model were utilized to study the specific surface area and pore size.

### 2.3. Electrochemical Performance Measurement

After mixing at 400 rpm for 12 h, the slurries of composites, polyacrylic acid (PAA, 10 wt.%), and Super P (Canard, Guangdong, China), with a mass ratio of 8:1:1, were pasted on the current collector as an electrode. During the slurry deposition, the doctor blade gap was 100 μm and the temperature of the coating machine was 40 °C. Then, the copper foil was used as the current collector and dried at a temperature of 90 °C for 10 h. Once the electrode had been dried to a constant weight, it was divided into circular discs with a mass loading of about 1.2 mg cm^−2^. The CR2025 coin-type cells were assembled in a glove box with an argon atmosphere, with a lithium metal plate (16 mm in diameter) as the counter electrode and a polypropylene membrane of Celgard 2400 as the separator; additionally, 1 M LiPF_6_ in a nonaqueous solution of ethyl methyl carbonate: ethylene carbonate: dimethyl carbonate (EMC:EC:DMC = 1:1:1 by volume) was used as the electrolyte. The CHI660E (Chenhua, Shanghai, China) workstation was used to carry on the cyclic voltammetry (CV, in a potential range of 0.01–1.50 V) test and electrochemical impedance spectroscopy (EIS, in a frequency range of 10^−2^–10^6^ Hz) test. The LANHE battery testing system (CT-2001, Wuhan, China) was used to test the electrochemical performance of cells and the galvanostatic intermittent titration technique (GITT, measured at 0.1 C (1 C = 2000 mAh g^−1^) for a 10 min pulse time between 1 h relaxation time) measurements. Before the half-cell tests, all the batteries were activated for three cycles at 0.1 C.

## 3. Results and Discussion

### 3.1. Characterization of Composites

The detailed synthesis process for the meso–macroporous multilayer silicon/carbon composite is delineated in Figure 1. First, the raw powder of silicon-cutting waste was dispersed in deionized (DI) water and stirred at high speed to obtain nano-sized silicon with uniform dimensions. As illustrated in Appendix A, the result of X-ray fluorescence spectrometry (XRF) analysis indicates that silicon is the primary constituent in pretreated silicon-cutting waste (SCW), but some heterogeneously distributed impurities, originating from residual organics, fragments of stainless steel, or loose diamond particles, were also detected. After high-speed stirring, the particle size became more uniform markedly; large particles were broken down into silicon nanoflakes with a D50 particle size of 180 nm (Appendix A), enhancing the subsequent coating uniformity and the electrical conductivity of the silicon particles. The pretreated SCW was oxidized in an air atmosphere. After annealing, residual organic materials in the SCW are eliminated, and the silicon particle surface is oxidized actively to form a complete SiO_2_ layer. Although the SiO_2_ coating layer sacrifices the silicon’s specific capacity, it improves the cycling performance of silicon anode substantially [29]. Finally, Si@SiO_2_ was incorporated into tetrahydrofuran (THF) with dissolved pitch and stirred until the THF was evaporated completely. The dry powders were then carbonized in a nitrogen atmosphere, generating double-layer coated Si/C composite materials. The aim of the 450 °C annealing is to transform isotropic pitch into anisotropic pitch to create a more oriented structure, which could accommodate the volume changes of silicon better [30]. This method has several advantages: (1) using inexpensive materials like SCW and pitch could reduce the cost of LIBs; and (2) providing a facile, easily scalable productive process for silicon anode.

The flake-like morphologies of materials were investigated by scanning electron microscopy (SEM). The Si@SiO_2_@C-3 (Figure 2b) is composed of silicon flakes of about 200 nm, retaining the morphology of SCW but having a reduced size (Figure 2a), and some pores after carbonization. These pores not only alleviate the volume expansions of silicon particles after the repeated cycles, enhancing the integrity of structure and cycle performance, but also promote the penetration of electrolyte, aiding the rate capability and transfer kinetics of Li^+^ [31,32]. The microstructure of Si@SiO_2_@C-3 was further investigated by high-resolution transmission electron microscopy (HRTEM), which includes the techniques of energy-dispersive X-ray spectroscopy (EDX) elemental analysis and selected-area electron diffraction (SAED). Figure 2c illustrates that the inner nanoflakes are coated with an oxidation layer and a carbon layer derived from the carbonized pitch, ensuring excellent electronic conductivity and facilitating strain relaxation, for this configuration enhances electron transfer kinetics and stabilizes the SEI layer [33]. Notably, HRTEM (Figure 2c) confirms the crystallinity of Si and graphite, with lattice spacings of 0.31 and 0.39 nm, respectively, associated with the (111) crystal plane of silicon and the (002) crystal plane of graphite. This is further corroborated by diffraction spots and rings of the SAED pattern, corresponding to the (111), (220), and (311) crystal planes of Si (Figure 2d). Additionally, EDX elemental mapping (Figure 2e–h) reveals that the distribution of Si and O is uniform across the center region, while C is primarily concentrated near the periphery, indicating that the exterior of the Si@SiO_2_ flake is coated with slightly graphitic carbon.

X-ray diffraction (XRD) patterns display the crystal phases of composites, as depicted in Figure 3a. Notably, all samples exhibited five similar peaks, at 28.3°, 47.2°, 56.2°, 69.1°, and 76.3°, which correlate with the (111), (220), (311), (400), and (331) crystal planes of silicon, confirming that the silicon retained high crystallinity during the process. Additionally, in the synthesized composites, a broad peak around 22.5° was identified, aligning with amorphous carbon, respectively. Combined with Figure 2c, the result suggests that the composites have disordered structures with a low degree of graphitization [34]. Figure 3b shows the Raman spectra for further analyzing the degree of graphitization in different Si@SiO_2_@C composites. Beyond a sharp peak of crystal silicon observed at approximately 518 cm^−1^, two broad carbon peaks (D and G bands) were detected at 1350 cm^−1^ and 1590 cm^−1^. As suggested in previous studies, the D band is associated with sp^3^ hybridization, reflecting the degree of disorder in the crystalline structure, and the G band corresponds to the first-order scattering of the E_2g_ vibrational mode [35]. Furthermore, the integral area values of D and G bands (*I*_D_/*I*_G_) serve as the metric to assess the graphitization degree in carbon-containing composites [36]. The decreased *I*_D_/*I*_G_ ratios of the Si@SiO_2_@C-2, 3 and 4 were estimated to be 0.86, 0.84, and 0.83, indicating an increasing degree of graphitization, which consequently enhances electrical conductivity and facilitates Li^+^ transport. The Si@SiO_2_@C-3 composite exhibits a typical type-IV N_2_ adsorption–desorption curve (Figure 3c) after coating of slightly graphitic carbon, indicative of meso–macroporous structures on its surface (Appendix A). These structures play a crucial role in alleviating morphological alterations due to substantial silicon volume expansion during discharging/charging cycles, thereby improving cycling stability [31,32,37]. Moreover, the Brunauer–Emmett-Teller (BET) analysis shows a decrease in specific surface area of composites from 114.49 (SCW) to 60.56 (Si@SiO_2_@C-3) m^2^ g^−1^ along with a reduction in pore volume (0.26 to 0.13 cm^3^ g^−1^) (Appendix A). This reduction is advantageous for minimizing side reactions, thus enhancing coulombic efficiency (CE) [14]. Thermogravimetric analysis (TGA, Figure 3d) determined the carbon content of Si@SiO_2_@C-3 to be 20.32 wt.%. For estimating the effect of carbon content, Si@SiO_2_@C-2 and Si@SiO_2_@C-4 were synthesized; they exhibit similar morphologies (Appendix A) although with different contents of carbon (16.26 wt.% for Si@SiO_2_@C-4 and 26.72 wt.% for Si@SiO_2_@C-2, Figure 3d).

The chemical bonds of Si, O and C in the Si@SiO_2_@C-3 composite were investigated by X-ray photoelectron spectroscopy (XPS). From the overall spectrum in Figure 4a, peaks corresponding to Si 2p and 2s, C 1s, and O 1s can be observed. As shown in Figure 4b, the Si 2p^1/2^, Si 2p^3/2^, Si-C and Si^4+^ were observed from Si 2p spectrum delineates, with four peaks at 98.7, 99.4, 100.7, 103.2 eV, confirming the presence of Si and SiO_2_ due to surface oxidation. Additionally, the C sp^2^, C sp^3^, -C-O, and -C=O bonds were identified from the C 1s spectrum in Figure 4c, with four peaks at 284.8, 285.8, 286.3, and 289.5 eV, respectively. The O 1s spectrum (Figure 4d) further corroborates the formation of a double-layer coating.

### 3.2. Electrochemical Performance of Electrodes

The evaluation of electrochemical performance was conducted with a coin-type half-cell system. Figure 5a illustrates cyclic voltammetry (CV) curves of the Si@SiO_2_@C-3 electrode at a scan rate of 0.1 mV s^−1^ with a voltage range of 0.01–1.50 V. In the initial cathodic scan, a wide peak appeared near 0.86 V, indicating that a series of irreversible reactions occurred between the electrolyte and the active materials, which are predominantly associated with the formation of the SEI layer. Notably, as the battery undergoes subsequent scans, this peak diminishes. An additional peak around 0.10 V is assigned to the alloying of Si into Li*_x_*Si, and two peaks of an initial anodic scan near 0.35 and 0.54 V are assigned to the dealloying of Li*_x_*Si [38]. Subsequently, a marked increase in the intensities of the two anodic peaks is observed, indicating the activation of the composite. Additionally, a cathode peak near 0.13 V, assigned to the lithiation process, appears in the latter cycles [39]. The CV curves of synthesized composites resemble that of SCW (Figure 5a and Appendix A), confirming the inherent electrochemical behavior of silicon-based materials. Figure 5b and Appendix A present the first-cycle discharge capacity and initial coulombic efficiency (ICE) of different electrodes. The first-cycle discharge specific capacity of SCW, Si@SiO_2_, Si@SiO_2_@C-4, Si@SiO_2_@C-3 and Si@SiO_2_@C-2 was 3085.9, 2485.9, 2050.1, 2044.6 and 1827.9 mAh g^−1^, correlating with an ICE of 53.01%, 49.59%, 87.01%, 84.12% and 83.99%, respectively. These results suggest that incorporating slightly graphitic carbon with a moderate content can reduce the formation of “dead lithium”, enhance electrode conductivity, and mitigate the volume change of silicon, thereby leading to an improved ICE of the multilayered composite. Figure 3c shows the result of the cycling test, conducted at 0.2 C for 100 cycles. It can be observed that the SCW with low electrical conductivity shows no measurable discharge/charge capacity. However, the integration of SiO_2_ and slightly graphitic carbon increases the residual capacity after 100 cycles substantially, which proves that the multilayered structure can significantly improve the Li^+^ storage capabilities of the SCW. Furthermore, it is noteworthy that the discharge capacity of Si@SiO_2_@C-4 begins to decline rapidly after approximately 90 cycles, indicating that insufficient carbon content does not effectively enhance the Li^+^ storage capacity. After 100 cycles, the Si@SiO_2_@C-3 notably maintains a higher capacity of 1549.6 mAh g^−1^ with an approximate coulombic efficiency of 99.5%, while Si@SiO_2_ maintains only 826.1 mAh g^−1^. Simultaneously, the coulombic efficiency of the third cycle achieves 95.5%, demonstrating the high reversibility of Li^+^ storage. This enhanced performance is attributable to the oxidation layer and protective carbon layer accommodating volume changes of silicon during discharging–charging cycles [31,32,33,37].

The rate performance (Figure 5d), is crucial for the practicality of materials for LIB electrodes. The Si@SiO_2_@C-3 retains an average discharge capacity of 973.5 mAh g^−1^ even at a rate of 2 C. Comparatively, the Si@SiO_2_@C-2 has no measurable capacity at 2 C, and the capacity of Si@SiO_2_@C-2 diminishes rapidly at 1 C and 2 C, demonstrating that the excess or deficiency of carbon content does not enhance rate performance correspondingly. Remarkably, the capacity of the electrode restores quickly to 1838.94 mAh g^−1^, as the current drops back down to 0.1 C. Notably, the Si@SiO_2_@C-3 composite outperforms other composites in both aspects of the cycling and rate performance significantly. Figure 5e depicts the 500-cycle cycling stability of the Si@SiO_2_ and Si@SiO_2_@C-3 at 0.5 C, with both maintaining a CE of approximately 99% after activation. However, after the long-term test, the residual capacity of the Si@SiO_2_@C-3 electrode is 711.4 mAh g^−1^, significantly higher than that of Si@SiO_2_. The results above indicate that the slightly graphitic carbon layer not only enhances the stabilization of the silicon nanoflake and avoids electrode pulverization, but also forms a stable SEI membrane and prevents the side reactions between silicon nanoflakes and electrolytes, thus ensuring outstanding cycling stability and the sustained high CE [40,41].

### 3.3. Interface Kinetics of Electrodes

The CV curves of the Si@SiO_2_@C composite at varying scan rates were recorded to study the lithium-ion storage mechanism in Figure 6a and Appendix A. Notably, the CV profiles increase correspondingly and maintain a consistent shape as the scan rates escalate, including one peak in cathodic scans (Peak 1) and two peaks in anodic scans (Peak 2 and 3). Furthermore, the currents (*i*) and the scan rates (*v*) adhere to Formula (1) [42]:(1)$$i=av^{b}$$
where *a* and *b* are the variates, and *b* can be calculated from the log (*i*)-log (*v*) plot depicted in Figure 6b. The *b* = 0.5 signifies that the Li^+^ storage behaviors of the electrode are controlled by the diffusion control behavior, and the *b* = 1.0 signifies that the Li^+^ storage behaviors of electrode are controlled by the capacitive control behavior, respectively. Figure 6b shows that the *b* of three peaks was fitted to be 0.76, 0.73 and 0.72, indicating that the interface kinetics of Si@SiO_2_@C-3 is controlled by a hybrid storage mechanism. The capacitance-controlled contributions can be calculated as per the Formula (2) [42]:(2)$$i\left(V\right)=k_{1}v+k_{2}v^{1/2}$$
where *k*_1_*v* and *k*_2_*v*^1/2^ stand for respective contributions of capacitive control behavior and diffusion control behavior. The *k*_1_ and *k*_2_ can be calculated by fitting. Figure 6c depicts the contributions of two interface kinetics. With the escalation of scan rates, the proportion of capacitance-controlled contribution either rises progressively (67%, 69%, 71%, 74% and 82% at 0.2, 0.4, 0.6, 0.8 and 1.0 mV s^−1^, respectively).

The electrochemical impedance spectroscopy (EIS, 10^−2^ to 10^6^ Hz) was conducted to investigate the diffusion kinetics of Li^+^ in carbon-coated composite electrodes further. The Nyquist plots of SCW and Si@SiO_2_@C electrodes were fitted in Figure 7a by the equivalent-circuit model. As depicted in EIS spectra, the solution resistance (*R_s_*) is represented at the point where the fitted curve intersects the Z′-axis. The semicircle at high frequencies corresponds to the charge–transfer resistance (*R_ct_*). The presence of a linear trend at low frequencies is indicative of the diffusion impedance of Li^+^, known as the Warburg impedance [43]. Appendix A shows the fitting values of *R_ct_* and Warburg of SCW and Si@SiO_2_@C electrodes. Compared with SCW (37.17 ohm), Si@SiO_2_@C-4 (57.19 ohm), Si@SiO_2_@C-3 (18.55 ohm), and Si@SiO_2_@C-2 (26.56 ohm), the Si@SiO_2_@C-3 electrode exhibits the lowest *R_ct_* (18.55 ohm), indicating that coating with an appropriate amount of carbon accelerates electron transmission. Furthermore, the following equations can be employed to estimate the Li^+^ diffusion coefficient (*D_Li_*^+^) [42]:(3)$$Z^{\prime }=R_{e}+R_{ct}+\sigma \omega^{-1/2}$$
(4)$$D_{{Li}^{+}}=\frac{R^{2}T^{2}}{2A^{2}n^{4}F^{4}C^{2}\sigma^{2}}$$
where ‘*σ*’ denotes the Warburg factor, ‘*A*’ signifies the electrode’s surface area, ‘*n*’ refers to the per unit charge transfer in the electrode reaction, ‘*F*’ is the Faraday constant, ‘*C*’ represents the Li^+^ molar concentration, ‘*R*’ stands for the gas constant, and ‘*T*’ indicates the absolute temperature. Figure 7b demonstrates that the fitted results of *σ* for the SCW, Si@SiO_2_@C-2, Si@SiO_2_@C-3 and Si@SiO_2_@C-4 electrodes are 236.16, 55.39, 32.11 and 69.26, respectively. The *D_Li_*^+^ of Si@SiO_2_@C-3 is 54.1 times higher than that of SCW, representing that the interface kinetics of the Si@SiO_2_@C-3 composite have been substantially optimized. The galvanostatic intermittent titration technique (GITT, Figure 7c) was used to calculate the *D_Li_*^+^ value by Fick’s second law [42]:(5)$$D_{{Li}^{+}}=\frac{4}{\pi \tau }{\left(\frac{m_{B}V_{M}}{M_{B}S}\right)}^{2}{\left(\frac{\Delta E_{S}}{\Delta E_{\tau }}\right)}^{2}$$
where *τ*, *M_B_*, *S*, *m_B_*, and *V_M_* stand for titration duration, molar mass, contact surface area of the electrode, active mass and molar volume. ∆*E_S_* and ∆*E_τ_* represent the potential variation between relaxation time and single step, respectively. As the potential changed, the *D_Li_*^+^ variation of the Si@SiO_2_@C-3 electrode exhibited a trend of the ‘W’-type (Figure 7d), due to the interactions between the lithium ions and composites, or the phase transitions from ordered to disordered [44,45]. Importantly, at the start of the discharge process, the *D_Li_*^+^ is relatively high, attributed to the slightly graphitic carbon, which facilitates the Li^+^ diffusion into the active materials [18]. The results demonstrate that the estimated *D_Li_*^+^ value of the Si@SiO_2_@C-3 electrode (10^−8^–10^−12^ cm^2^ s^−1^) is more than 100 times higher than the SCW electrode [44,45], implying the significantly enhanced Li^+^ diffusion kinetics of the electrode. From the preceding discussion of the Li^+^ diffusion kinetics, the optimized rate and cycling performance of Si@SiO_2_@C-3 could be attributed to its distinctive multilayered structure: (1) the porous double-layered structure can alleviate the repeated volume expansions of silicon; (2) the porous structure can facilitate lithium ion and electron transport, resulting in the rapid diffusion kinetics of Li^+^ throughout the discharge–charge cycles; (3) the coating of high electro-conductive slightly graphitic carbon significantly improves Li^+^ diffusion kinetics across interfaces. Therefore, the integration of carbon derived from pitch not only improves the Li^+^ storage capability of silicon-derived electrodes but also serves as a viable strategy to endow other electrodes with the enhanced integrity of structure and the accelerated interface kinetics of Li^+^.

As shown in Figure 8, SEM and quantitative analyses of Si@SiO_2_ and Si@SiO_2_@C-3 composites were used to elucidate the influence of a carbon layer on the integrity of electrode morphologies. Prior to the cycling test, the electrode surfaces are almost identically flat (Figure 8a,c). However, the morphologies of the electrodes are obviously different after long-term cycles. With no slightly graphitic carbon layer, the Si@SiO_2_ electrode struggles to accommodate the tremendous stress of volume expansion, and numerous large cracks appear on the Si@SiO_2_ electrode; some parts of the active materials even lose contact with the copper foil. The Si@SiO_2_ electrode exhibits a rise in thickness from 13.1 to 46.2 μm with a volumetric expansion rate of 252.7% (Figure 8a,b). Quite the opposite, the Si@SiO_2_@C-3 electrode maintains a flat and intact morphology after 500 cycles, with only a few fine cracks. The Si@SiO_2_@C-3 electrode grows from 11.8 μm to 34.1 μm in thickness with a relatively smaller volumetric expansion rate of 188.9% (Figure 8c,d), due to the contribution of a slightly graphitic carbon layer that effectively alleviates the stress of volume expansion.

### 3.4. Application in Full Cells

Based on the outstanding performance exhibited by Si@SiO_2_@C-3, a full cell, with Li[Ni_0.8_Co_0.1_Mn_0.1_]O_2_ (NCM811) as the cathode and Si@SiO_2_@C-3 as the anode, was constructed to assess its practical suitability. Before assembling, the Si@SiO_2_@C-3 was prelithiated by a half-cell system at 0.1 C for three cycles. Figure 9a shows the mechanism of the discharging process of full cells; during the discharging process, the electrons and ions move from the anode to the cathode through different paths. The light-emitting diode (LED) was lit up by the action of electron flow, as shown in Figure 9b. The GCD curve in Figure 9c illustrates that the full cell has a significant first-cycle capacity of 229.4 mAh g^−1^, with an ICE of 73.4%. Furthermore, the full cell has a capacity retention of 71.2% at 0.2 C after 50 cycles, and an energy density of 493.7 Wh kg^−1^ (Figure 9d). The above-mentioned exceptional full-cell performance demonstrates the future practical application potential of synthesized multilayered porous composite materials for LIBs.

### 3.5. Comparison of This Work to the State of the Art

Reutilizing the low-value waste generated from silicon ingot cutting by transforming it into a high-performance anode is highly beneficial for sustainable development and environmental protection. In recent years, several studies have explored the use of silicon-cutting waste as a low-cost raw material for the fabrication of various silicon–carbon anode materials [46,47,48,49,50,51]. In this work, a facile and cost-effective method is designed for constructing anodes with outstanding performance for LIBs. The comparison of this work to the state of the art in the method and electrochemical performance of anodes derived from silicon-cutting waste is shown in Table 1. It is noteworthy that the annealing method is safer and more environment-friendly, for hydrothermal reactions [46,48] and chemical vapor deposition (CVD) [47] need high-temperature, high-pressure conditions, and CVD also contains unstable deposited gases. The chemical etching method [49,50] could generate porous silicon-based composites with enhanced electrochemical performance, but the generation of H_2_ and strong acid/base waste liquid during the synthesis process leads to numerous environmental issues. Compared to silicon–carbon anodes, the composite anodes with well-conductive precious metals, such as silver (Ag) [50], show better electroconductivity indeed, but impose higher production costs as well. On the selection of carbon source, the use of cost-effective coal tar pitch further reduces the cost of synthesis. Thus, the method adopted in this work and the electrochemical performance of the obtained multilayer Si@SiO_2_@C composite show certain superiorities.

## 4. Conclusions

In summary, a facile and economical synthesis technique has been designed to transform silicon-cutting waste into high-performance anodes for LIBs with a double-layered porous structure. The porous structure of the obtained Si@SiO_2_@C composite effectively mitigates the repetitive and substantial volume expansion of silicon during discharging/charging cycles. Simultaneously, the silicon nanoflakes are coated with both an oxide layer and a carbon layer, which enhance the electrical conductivity and mechanical strength of the electrode, thus contributing to the migration of the ion/electron and the integrity of electrodes significantly. With the preceding benefits, the Si@SiO_2_@C-3 composite shows exceptional electrochemical performance, including an impressive high initial discharge capacity of 1994.8 mAh g^−1^ at 0.2 C with a high ICE of 84.12%, remarkable cycle stability with 711.4 mAh g^−1^ residual capacity at 0.5 C after 500 cycles, and exceptional rate capability of 973.5 mAh g^−1^ up to 2 C. Consequently, this research not only introduces a cost-effective technology for synthesizing silicon-based anodes for high-energy-density LIBs but also opens green strategies for the sustainable resource utilization of silicon-cutting waste.

## Figures and Tables

**Figure 1 nanomaterials-14-00625-f001:**
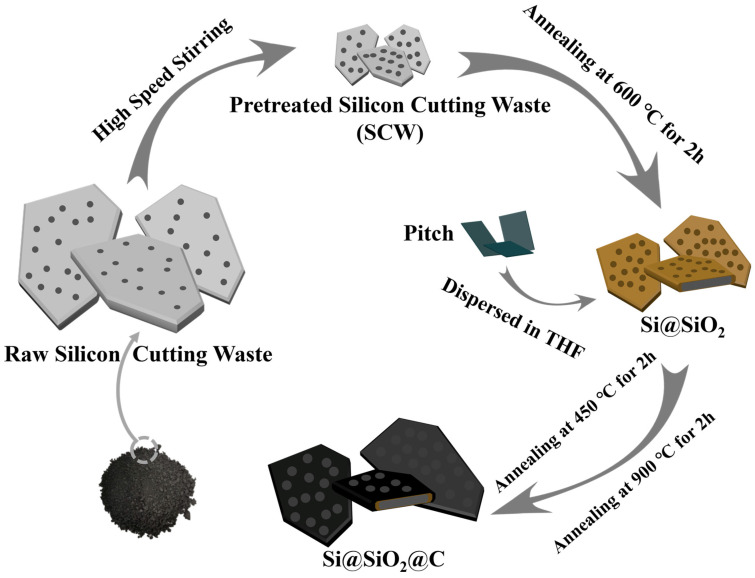
The formation process of Si@SiO_2_@C.

**Figure 2 nanomaterials-14-00625-f002:**
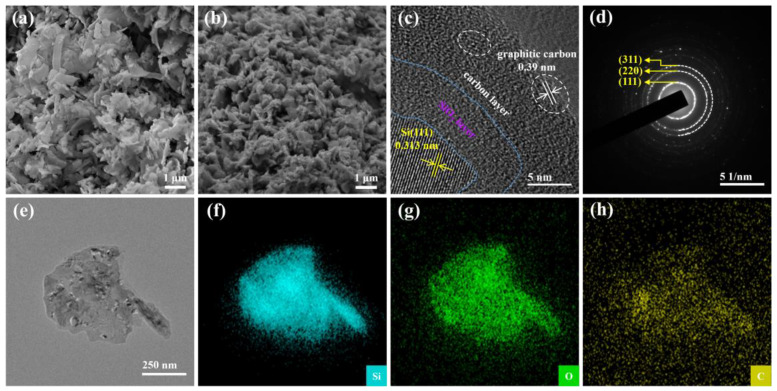
(**a**,**b**) Surface morphologies of SCW and Si@SiO_2_@C-3, (**c**) high-resolution TEM, (**d**) SAED and (**e**–**h**) elemental mapping patterns of Si@SiO_2_@C-3.

**Figure 3 nanomaterials-14-00625-f003:**
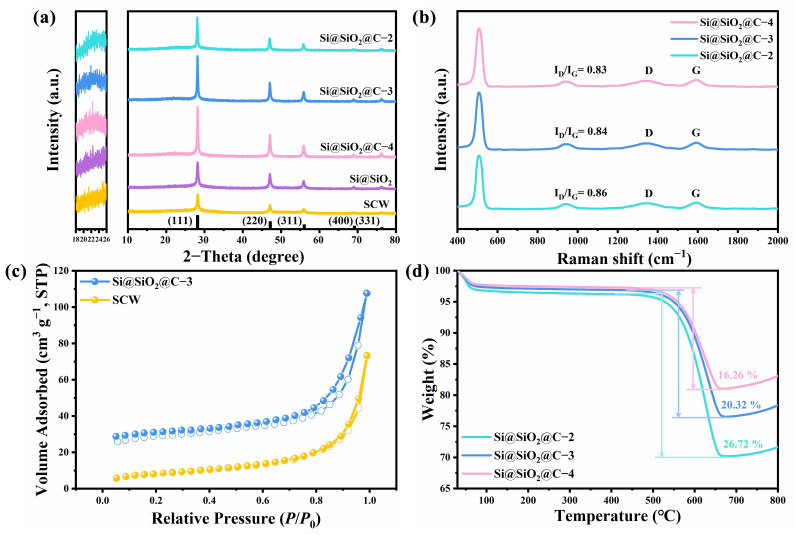
(**a**) XRD patterns, (**b**) Raman spectra, (**c**) nitrogen sorption isotherms, and (**d**) TGA profiles.

**Figure 4 nanomaterials-14-00625-f004:**
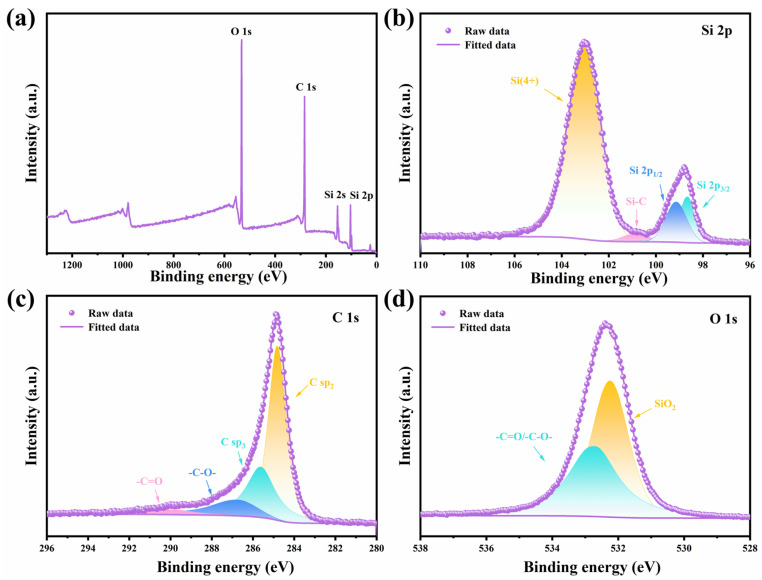
(**a**–**d**) High-definition XPS spectrum of Si@SiO_2_@C-3.

**Figure 5 nanomaterials-14-00625-f005:**
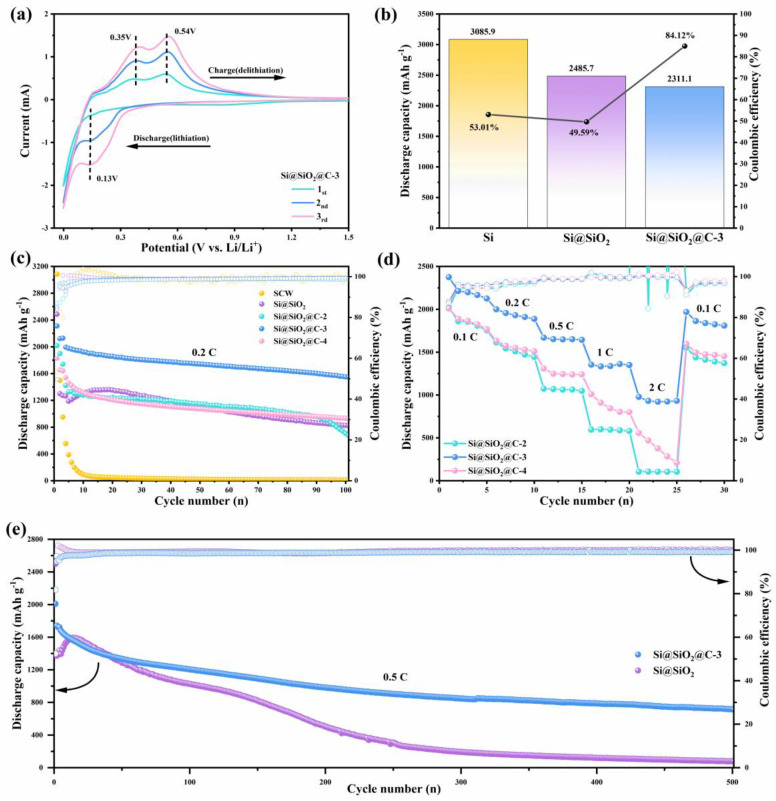
(**a**) CV curves at 0.1 mV s^−1^, (**b**) 1st discharge capacity and initial coulombic efficiency, (**c**) cycling performance at 0.2 C for 100 cycles, (**d**) rate capability at 0.1 C, 0.2 C, 0.5 C, 1 C and 2 C, and (**e**) long-term cycling stability at 0.5 C for 500 cycles.

**Figure 6 nanomaterials-14-00625-f006:**
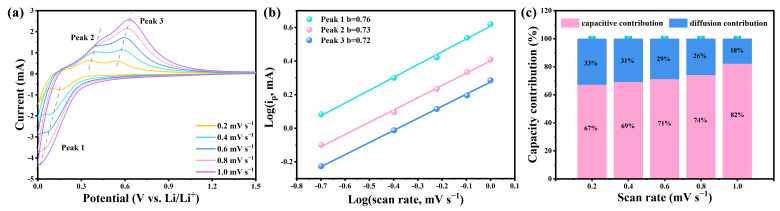
(**a**) The CV curves, (**b**) log(*i*)/log(*v*) plots and (**c**) capacitance-controlled contribution for Si@SiO_2_@C-3 at different scan rates.

**Figure 7 nanomaterials-14-00625-f007:**
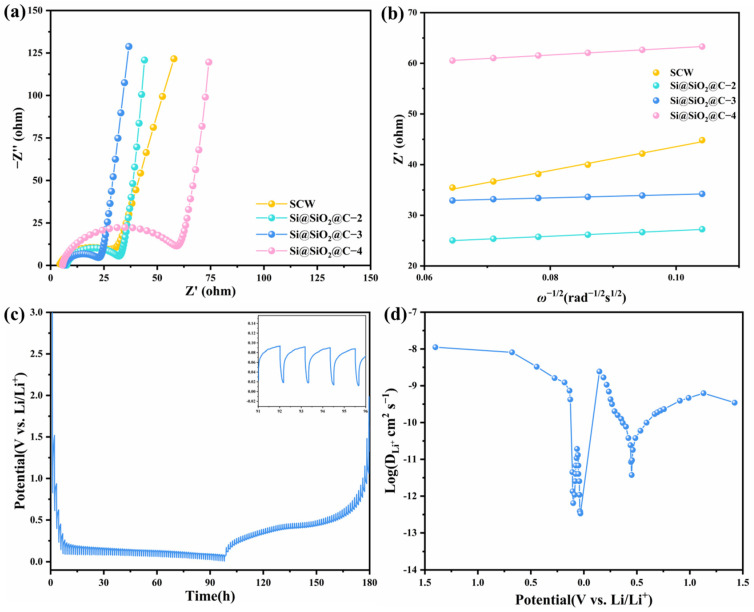
(**a**) EIS of electrodes, (**b**) *Z*′/*ω*^−1/2^ plot, (**c**) GITT curves and (**d**) Li^+^ diffusion coefficients of Si@SiO_2_@C-3.

**Figure 8 nanomaterials-14-00625-f008:**
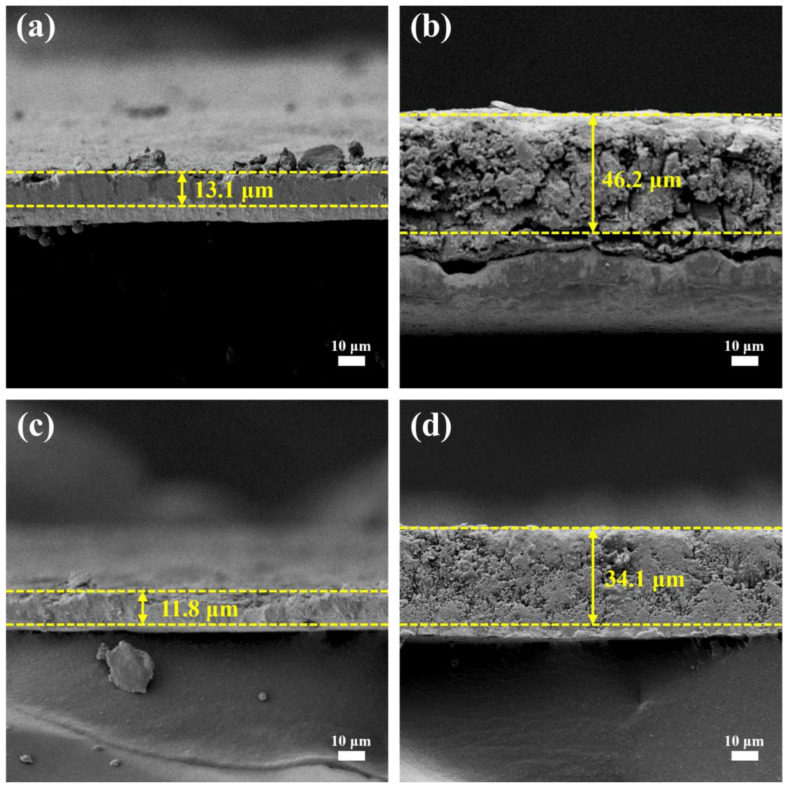
Morphological evolution of (**a**,**b**) Si@SiO_2_ and (**c**,**d**) Si@SiO_2_@C-3 electrodes before and after 500 cycles.

**Figure 9 nanomaterials-14-00625-f009:**
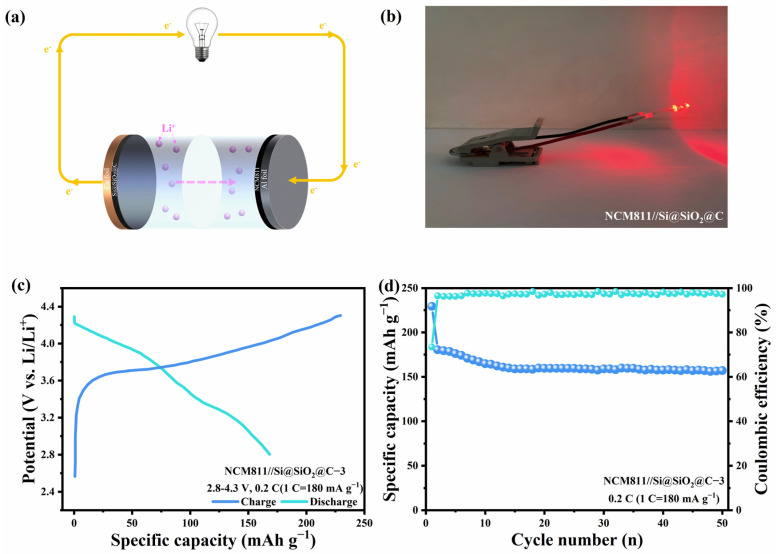
(**a**) Schematic illustration of discharging process, (**b**) images of LED lit by the full cell, (**c**) initial GCD curve at 0.1 C, (**d**) cycling performance at 0.2 C for 50 cycles.

**Table 1 nanomaterials-14-00625-t001:** Comparison of this work to the state of the art in method and electrochemical performance of anodes derived from silicon-cutting waste. (1 C = 2 A g^−1^).

Materials	Method	Carbon Source	Rate Capability	Cycle Performance	Ref.
FP-Si@C-2	Hydrothermal reaction, Annealing	Excrementum Bombycis	1 A g^−1^, 406 mAh g^−1^	0.1 A g^−1^, 100 cycles, 935.9 mAh g^−1^	[46]
Si/CNTs950-1.5	CVD (acetylene)	Carbon nanotubes	2 A g^−1^, 619.8 mAh g^−1^	1 A g^−1^, 100 cycles, 920.1 mAh g^−1^	[47]
Si/CTS/G	Hydrothermal reaction, Annealing	Chitosan/Graphite	1 A g^−1^, 487 mAh g^−1^	0.1 A g^−1^, 400 cycles, 750 mAh g^−1^	[48]
Si@PC	Self-assembly, Annealing and HCl-assisted pickling	Citric acid	1 A g^−1^, 505.4 mAh g^−1^	0.5 A g^−1^, 200 cycles, 712.6 mAh g^−1^	[49]
pSi/Ag/C/G	Silver-assisted chemical etching, Annealing	PVP/Graphite	1 A g^−1^, 1410 mAh g^−1^	1 A g^−1^, 200 cycles, 971 mAh g^−1^	[50]
Si@NC-ZIF	In situ grown, Annealing	CTAB	2 A g^−1^, 611.7 mAh g^−1^	0.2 A g^−1^, 100 cycles, 1623.05 mAh g^−1^	[51]
**Si@SiO_2_@C-3**	**Two-step** **annealing**	**Pitch**	**2 C,** **973.5 mAh g^−1^**	**0.2 C, 100 cycles,** **1549.6 mAh g^−1^**	**This work**
				**0.5 C, 500 cycles,** **711.4 mAh g^−1^**	

## Data Availability

Data are contained within the article and Appendix A.

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
