# Peer review of "Reutilization of Silicon-Cutting Waste via Constructing Multilayer Si@SiO2@C Composites as Anode Materials for Li-Ion Batteries"

_nanomaterials, 2024, doi:10.3390/nano14070625_

Round 1
Reviewer 1 Report
Comments and Suggestions for Authors
The study proposes a method to reuse silicon waste from photovoltaic industry by creating a composite for Li-ion batteries. The composite's structure effectively handles silicon volume changes and enhances conductivity, resulting in improved battery performance. This offers a cost-effective way to repurpose silicon waste into high-energy-density battery materials. Manuscript mechanics is good, discussions are adequate and conclusions justified. Thus, this manuscript can be accepted for publication after minor revision taking into account next points:
Line 81: “in to” should be replaced by “into”
Line 111: a detailed methodology of the slurry preparation should be explained. Moreover, authors should incorporate the doctor blade gap and the temperature employed during the slurry deposition.
Line 145: The synthesis of Si@SiO2@C has already been described in line 86, so the experimental data described in this paragraph appear to be redundant.
The figures do not appear in the same order as referenced in the text. For example, in Figure 3, the text first explains Figure 3d and then Figure 3a. Then, Figure 2c is explained. The figures should appear in the same order as referenced in the text to facilitate reading. Please make the change accordingly.
Line 303: the equivalent-circuit diagram for the fitted EIS curves should be included in Figure 7a as inset instead of in Figure S7.
Comments on the Quality of English Language
Line 81: “in to” should be replaced by “into”
Author Response
Cover Letter
Dear Reviewer,
On behalf of my co-authors, I’d like to thank you very much for giving us an opportunity to revise our manuscript. These comments and suggestions are all valuable and very helpful for revising and improving our manuscript (nanomaterials-2936256). According to your suggestions, all the points have been revised and the point-by-point responses have been listed after this cover letter. The attachment is the revised manuscript.
We have studied the comments carefully and tried our best to revise our manuscript according to the comments, all the revisions were marked in yellow in the revised manuscript.
We would like to express our great appreciation to you for the valuable comments and suggestions again and we are looking forward to hearing from you.
Best regards,
Yours sincerely,
Chunyan Lai
Shanghai Key Laboratory of Materials Protection and Advanced Materials in Electric
Power, College of Environmental and Chemical Engineering, Shanghai University of
Electric Power
E-mail address: laichunyan@shiep.edu.cn
Reply to Reviewer #1:
Comments:
The study proposes a method to reuse silicon waste from photovoltaic industry by creating a composite for Li-ion batteries. The composite's structure effectively handles silicon volume changes and enhances conductivity, resulting in improved battery performance. This offers a cost-effective way to repurpose silicon waste into high-energy-density battery materials. Manuscript mechanics is good, discussions are adequate and conclusions justified. Thus, this manuscript can be accepted for publication after minor revision taking into account next points:
- Line 81: “in to” should be replaced by “into”
Response: Thank you very much for your comments. The errors in English grammar have been revised on Line 81.
Action: “in to” has be replaced by “into” on Line 81.
Relevant changes have been added to the main text and marked in yellow.
- Line 111: a detailed methodology of the slurry preparation should be explained. Moreover, authors should incorporate the doctor blade gap and the temperature employed during the slurry deposition.
Response: Thank you so much for your constructive comments. The rate and time of the slurry mixing has been added on Line 111, and the doctor blade gap and the temperature employed during the slurry deposition has been detailed on Line 113.
Action: A detailed methodology of the slurry preparation has been explained: “After mixing at 400 rpm for 12 h, the slurries of composites, polyacrylic acid (PAA, 10 wt%), and Super P (Canard, China), with a mass ratio of 8: 1: 1, were pasted on the current collector as an electrode. During the slurry deposition, the doctor blade gap was 100 μm and the temperature of the coating machine was 40 ℃.”
Relevant changes have been added in the main text and marked in yellow.
- Line 145: The synthesis of Si@SiO2@C has already been described in line 86, so the experimental data described in this paragraph appear to be redundant.
Response: Thank you very much for your constructive suggestion. The redundant data in this paragraph has been deleted.
Action: Relevant changes have been added to the main text and marked in yellow.
- The figures do not appear in the same order as referenced in the text. For example, in Figure 3, the text first explains Figure 3d and then Figure 3a. Then, Figure 2c is explained. The figures should appear in the same order as referenced in the text to facilitate reading. Please make the change accordingly.
Response: Thank you so much for your comments. The order of all the figures has been rearranged, and all the figures appears after the corresponding text after the change.
Action: Relevant changes have been added to the main text and marked in yellow.
- Line 303: the equivalent-circuit diagram for the fitted EIS curves should be included in Figure 7a as inset instead of in Figure S7.
Response: Thank you very much for your constructive comments. The equivalent-circuit diagram for the fitted EIS curves has been added in the upper-right corner of Figure 7a.
Action: Relevant changes have been added to the main text and marked in yellow.
Reviewer 2 Report
Comments and Suggestions for Authors
Dear Authors,
You did a good and important work. It has obvious technical and scientific value. My detailed review with technical remarks is attached.
Only several grammar errors should be eliminated. After the final editing, it can be published.
Good lucks,
the reviewer

Comments on the Quality of English Language
Only some of the grammar errors pointed out in the review need to be eliminated.
Author Response
Cover Letter
Dear Reviewer,
On behalf of my co-authors, I’d like to thank you very much for giving us an opportunity to revise our manuscript. These comments and suggestions are all valuable and very helpful for revising and improving our manuscript (nanomaterials-2936256). According to your suggestion, all the points have been revised and the point-by-point responses have been listed after this cover letter.
We have studied the comments carefully and tried our best to revise our manuscript according to the comments, all the revisions were marked in red in the revised manuscript.The attachment is the revised manuscript.
We would like to express our great appreciation to you for the valuable comments and suggestions again and we are looking forward to hearing from you.
Best regards,
Yours sincerely,
Chunyan Lai
Shanghai Key Laboratory of Materials Protection and Advanced Materials in Electric
Power, College of Environmental and Chemical Engineering, Shanghai University of
Electric Power
E-mail address: laichunyan@shiep.edu.cn
Reply to Reviewer #2:
Comments:
The presented article is dedicated to the solution of a clever utilization of silicon cutting waste produced during the technology process of PV cell production. Solving the silicon waste problem is extremely important due to the significant amounts of silicon powder, nanoparticles, and nanoflakes that remain during cell fabrication, increasing production costs and environmental pollution. In this perspective, the presented work is scientifically and technically significant. The manuscript is written in a clear and understandable manner. However, several grammar errors do not spoil the overall impression and can be eliminated during final editing.
- In Abstract L.17: The sentence should be as follows: The rapid development of the photovoltaic industry has also brought some economic losses and environmental problems due to the waste generated during silicon ingot cutting.
Please, eliminate the sentence errors in accordance.
1.1 L.43, should be: “…silicon-cutting…”
1.2 L.50: should be: “…of over…”
1.3 L.81: should be “…into a container…”
1.4 L.102: should be “…Raman spectroscopy…”
1.5 L.103: “delete And…”
1.6 L.111: should be “…Canard…”
1.7 L.116: to add article a – “…a lithium metal plate…”
1.8 L.118: once again the lack of “a” – “…in a (or the) nonaqueous solution…”
1.9. L.167: should be “…from the carbonized pitch…”
1.10. L.189: “…with the (to delete!) Figure 2c…”
1.11. L190: “…with a low…”
1.12. L.224: should be “…reactions occurred…”
1.13. L.241: should be: “…of the cycling test,…”
1.14. L.272: “…thus ensuring the (to delete) outstanding cycling stability…”
1.15. L.285: “…of the (to add) electrode are…”
1.16. L.292: “…for respective contributions…”
1.17. L.304: “…at high frequencies correspond(s)…”
1.18. L.330: “…start of the (to add) discharge process,…”
1.19. L.334: “…kinetics (to add s) of the (to add) electrode…”
1.20. L.359: “…(Figures (to add) 8c and d),…”
1.21. L.360: “…of a(to add) slightly graphitic carbon…”
1.22. L.369: “…by a(to add) half-cell system…”
1.23. L.371: “…from the (to add) anode…”
1.24. L.380: “…into a(to add) high-performance anode…”
1.25. L.382: “…of(to add) silicon cutting waste as a low-cost raw material for the
fabrication of(to add) various silicon-carbon anode…”
1.26. L.383: “And (to delete) in this work…”
1.27. L.389: “…unstable deposited gases…”
1.28. L.397: “…composite shows…”
1.29. L.405: “…with both an….”
1.30. L.410: “…with a(to add) high ICE….”
1.31. L.412: “…not only introduces(to add)…”
Response: Thank you very much for your constructive comments. All the errors in the English grammar mentioned above have been revised in the manuscript.
Action: Relevant changes have been added to the main text and marked in red.
- The introduction should be added as the final section, providing a brief overview of the work's structure.
Response: Thank you very much for your comments. The brief overview of the work’s structure has been further supplemented in the last paragraph of the introduction.
Action: Add the brief overview of “3.4. Application in Full Cells” and “3.5. Comparison of This Work to the State of the Art” in the last paragraph of the introduction to make the introduction more complete.
Relevant discussions have been added to the main text and marked in red.